# The Effect of Antiglaucoma Procedures (Trabeculectomy vs. Ex-PRESS Glaucoma Drainage Implant) on the Corneal Biomechanical Properties

**DOI:** 10.3390/jcm10040802

**Published:** 2021-02-17

**Authors:** Aristeidis Konstantinidis, Eirini-Kanella Panagiotopoulou, Georgios D. Panos, Haris Sideroudi, Aysel Mehmet, Georgios Labiris

**Affiliations:** Department of Ophthalmology, University Hospital of Alexandroupolis, 68131 Alexandroupolis, Greece; eipanagi@med.duth.gr (E.-K.P.); gdpanos@gmail.com (G.D.P.); hsideroudi@gmail.com (H.S.); ayselmehmet83@hotmail.com (A.M.); labiris@usa.net (G.L.)

**Keywords:** glaucoma, trabeculectomy, corneal hysteresis, corneal resistance factor

## Abstract

The aim of this study is to investigate the effect of two antiglaucoma procedures, namely trabeculectomy and Ex-PRESS mini-shunt insertion on the biomechanical properties of the cornea. This is a prospective study. Thirty patients (30 eyes) were included in the study. Nineteen eyes had an Ex-PRESS shunt inserted (Group 1) and 11 had trabeculectomy (Group 2). The examination time points for both groups were one to three weeks preoperatively and at month 1, 6, and 12 postoperatively. Corneal biomechanical properties (corneal hysteresis (CH) corneal resistance factor (CRF)) were measured with the Ocular Response Analyzer (ORA). In group 1, CH was significantly increased at 6 and 12 months compared to baseline values. Corneal hysteresis was also higher at 1 month postoperatively, but this increase did not reach statistical significance. In group 2, the CH was significantly increased at all time points compared to the preoperative values. CRF decreased at all time points postoperatively compared to the preoperative values in both groups. The difference (preoperative values to postoperative values at all time points) of the CH and CRF between the two groups was also compared and no significant differences were detected between the two surgical techniques. Trabeculectomy and the EX-PRESS mini-shunt insertion significantly alter the corneal biomechanical properties as a result of the surgical trauma and the presence of the shunt in the corneal periphery. When compared between them, they affect the corneal biomechanical properties in a similar way.

## 1. Introduction

Glaucoma is the second cause of blindness globally after cataract [1]. However unlike cataract it can cause irreversible loss of vision. Intraocular pressure (IOP) is a major risk factor for glaucomatous optic neuropathy and the only one that can be modified [2]. The initial treatment of the chronic forms of glaucoma is the conservative management with eye drops. In many cases, the drop of the IOP is not sufficient to slow down the optic nerve damage and the various surgical options are explored.

Trabeculectomy (TM) has been the standard surgical approach since its first description by Cairns [3]. Although it is a successful operation in terms of IOP control [4,5], it has been associated with a considerable number of early and late complications [6] and this has given rise to the development of newer techniques with less complications. The Ex-PRESS glaucoma implant, on the other hand, is made of stainless steel and does not have an internal valve mechanism.

The measurement of the IOP with the Goldmann applanation tonometer (GAT) is the standard clinical practice despite the existence of numerous other tonometers. The optimal area of applanation is based on the Imbert-Fick principle and assumes that the cornea is perfectly elastic, infinitely thin, and dry [7]. As the cornea has none of these features, the accuracy of the measurements with the GAT is limited [7].

Research has shown that the cornea is a more complex structure than a simple elastic surface but also has viscous properties that make the cornea a perplex viscoelastic tissue [8]. The influence of these properties is far higher than the influence of the central cornea thickness and curvature [9]. The Ocular Response Analyzer (ORA; AMETEK Inc. and Reichert Inc., Depew, NY, USA) is a device that can measure both the IOP and the biomechanical properties of the cornea.

The principle of its function relies on the emission of a precisely metered air pulse of 20 ms duration [10]. The ORA has a coupled infrared transmitter, which radiates infrared light on the cornea. This light is reflected by the corneal surface and is detected by the receiver. When the pressure on the cornea by the air jet is such that it applanates the central 3 mm of the corneal surface, then the detected infrared light intensity is maximum and this point corresponds to the inward applanation (P1). The pressure of the air jet continues to increase for a few more milliseconds and then gradually decreases until the cornea becomes flat again due to its elastic properties. At this point, the infrared light intensity is maximum again and the instrument measures the outward applanation pressure (P2). The ORA uses two indices to measure the corneal biomechanical properties: (i) The corneal hysteresis (CH), which is a measure of the viscoelastic properties, and (ii) the corneal resistance factor (CRF), which sums up the effects of the corneal material properties, central corneal thickness and curvature [11]. The CH is calculated as P1–P2 and the CRF is derived from the equation P1–kP2, where k is a constant. When the constant is calculated at *k* = 0.68, then the CRF has its maximum association with the central corneal thickness (CCT) [12]. The ORA offers two different methods of measuring the IOP. The IOP(g) is the average of P1 and P2. The IOP(cc) is calculated from the equation P2-kP1 where k is a constant, which when given the value of 0.43, it is the least dependent on the corneal thickness [12.Within this context, the primary objective of this study was to assess whether the two antiglaucoma procedures (TM and the insertion of the Ex-PRESS mini shunt) affect the biomechanical properties of the cornea differently.

## 2. Experimental Section

### 2.1. Setting

This was a prospective, comparative study. Study protocol adhered to the tenets of the Helsinki Declaration and written informed consent was obtained by all participants. The institutional review board of the Democritus University of Thrace approved the protocol (ethical approval code: ES8/Th11/10-10-2013) and study was conducted at the University Hospital of Alexandroupolis (UHA), in Greece between July 2013 and May 2016. Official registration number of the study is NCT04648943.

### 2.2. Participants

Participants were recruited from the Glaucoma Service of the UHA in a consecutive-if-eligible basis and populated two distinct groups for the purposes of this study: (i) Group 1: Eyes having an Ex-PRESS shunt inserted, and (ii) Group 2: Eyes that underwent TM. Exclusion criteria for both groups included previous ocular trauma, ocular surgery other than phacoemulsification, previous disease of the ocular surface, and congenital glaucoma.

The examination time points for both groups were 1 to 3 weeks preoperatively and 1 month, 6 months, and 12 months postoperatively. All patients had a thorough ophthalmic examination at all time points and the corneal biomechanical properties were measured with the ORA by a trained technician. The ORA measurements were taken before the instillation of the anesthetic drops for the IOP measurement with the GAT. Measurements with a Waveform Score ≤ 3.5 were excluded.

### 2.3. Surgical Technique-Postoperative Management

All operations were performed by two experienced surgeons (VK, AK) in a consistent way. The surgical technique for the TM was as follows: A conjunctival peritomy was done at the limbus with blunt dissection of the conjunctiva/tenon’s capsule. Unipolar cautery was kept to a minimum. A 4 × 4 mm limbus-based flap was formed at roughly half the sclera thickness. Mitomycin C (MMC 0.2 mg/mL for 2–3 min) was applied with the use of a few pieces of a Weck-cell sponge arranged over a wide area under the conjunctiva. The edges of the conjunctiva were grasped with serrated forceps and were wiped with clean Weck-cell sponges. The area of application of MMC was then irrigated with 20 mL of balanced salt solution in order to wash away the MMC. A side port was created with a 20 G knife. Two 10/0 Nylon sutures were preplaced at the corners of the scleral flap. The anterior chamber was entered under the flap with a 45° knife. A corneoscleral block of tissue was excised with a Kelly punch in order to create the internal ostium. A peripheral iridotomy was performed with scissors. More 10/0 Nylon sutures may be placed to the flap according to the surgeon’s discretion. The conjunctiva was closed with 2 to 4 10/0 Nylon sutures. Dispersive viscoelastic was injected under the conjunctiva to create a space between the conjunctiva and the sclera for the first postoperative days. A long-acting solution of betamethasone was injected under the conjunctiva behind the filtering bleb and intracameral antibiotic was also used.

There are several models of the implant but in the current study the P50 model was used. It has an internal diameter of 50μm, an external diameter of 0.4 mm, and is 2.46 mm long. The use of this mini shunt has shown to be as effective as the standard TM with fewer side effects [13,14]. The surgical technique for the insertion of the Ex-PRESS mini shunt was the same with the only differences being that the anterior chamber was entered under the flap with a 25 G needle at the anterior part of the blue transition zone (which internally corresponds to the trabeculum). The mini shunt was inserted through the track created by the needle. A peripheral iridotomy was not required.

Topical steroids were prescribed every 2 h tapered gradually according to the surgeon’s discretion. Topical antibiotics were prescribed 4 times/day for 1 month. Topical cyclopentolate 1% was given to the trabeculectomy group but not to the Ex-PRESS group as the postoperative inflammation in the latter group is minimal.

The patients were examined in the first postoperative day and at the predetermined time points as mentioned above. At each time point, the surgeon injected dexamethasone with or without 5-fluorouracil and performed needling to the filtering bleb according to his discretion.

### 2.4. Data Collection

All parameters were measured before surgery and at 1, 6, and 12 months after surgery. The primary outcome measures were the CRF and the CH measured with the use of the ORA, while the secondary outcome measure was the IOP measured with the GAT.

### 2.5. Statistical Analysis

Medcalc software version 18.2.1 (MedCalc Software bvba, Ostend, Belgium) was used for the statistical analysis. Data distribution was tested with Shapiro–Wilk test and Q-Q plot and parametric and non-parametric tests were applied accordingly. Data are presented as mean ± standard deviation (SD) or error (SE) when the distribution was normal or as median (minimum-maximum) when the distribution was skewed. The power of all statistical tests used was greater than 0.8, suggesting that the size of our sample was sufficient (G*Power 3.1.9.2, University of Dusseldorf, Dusseldorf, Germany). *p*-values < 0.05 were defined as statistically significant.

## 3. Results

Thirty patients (30 eyes) were included in the study. Nineteen eyes had an Ex-PRESS shunt inserted (Group 1) and 11 eyes underwent TM (Group 2). Detailed demographic and clinical parameters of each group are presented in Table 1. Non-significant differences were detected with respect to age (*p* = 0.18). No ocular parameter demonstrated significant differences between the two groups preoperatively (*p* values: 0.2 to 0.43).

In group 1, the data were analyzed with parametric indices using the ANOVA test. The CH was significantly increased at 6 and 12-month time points compared to baseline values (Table 2). CH was also higher at 1 month postoperatively, but this increase did not reach statistical significance. The mean CH was 7.31 preoperatively and increased to 7.92 at 1 month to 8.32 at 6 months and 8.37 at 12 months. On the other hand, CRF was decreased significantly at all time points postoperatively compared to the preoperative values (Table 3). The mean preoperative CRF was 10.6, at 1 month it was 8.07, at 6 months 8.12, and at 12 months 8.35.

In group 2, the CH the data were analyzed with non-parametric indices due to the small sample using the Friedman test. The CH was significantly increased at all time points compared to the preoperative values (Table 4). The median preoperative CH was 7.85, at 1 month 8.9, at 6 months 8.8, and at 12 months 8.6. The CRF in the same group was analyzed with the ANOVA test for parametric data. The mean preoperative CRF was 11.11 and it was decreased to 8.21 at the first postoperative month, 8.33 at 6 months, and 8.29 at 1 year. The decrease of the CRF was significant at all time points compared to the preoperative values (Table 5).

We performed a regression analysis to check for any correlation of the CH and CRF change taking into account the IOP change. With GAT-IOP as a covariate, we found that the CH and CRF change (before and after surgery) in both groups was not significant at all time points (all *p* < 0.05).

We also carried out multivariate analysis using a linear regression model, with a stepwise backward elimination procedure including changes in IOP, CH, CRF, and patients’ age in order to find any correlation between these variables. Multivariate analysis did not reveal any correlation between changes in biomechanical properties (CH, CRF) and patients’ age and/or IOP changes.

Finally, we compared the difference (preoperative values to postoperative values at all time points) of the CH between the two groups (Welch test) and we did not detect any significant differences between the two surgical techniques (Table 6). When the CRF changes (preoperative values to postoperative values at all time points) were compared between the two groups (using the t-test), they were not found to be significant (Table 7).

The IOP was significantly reduced at all postoperative time points with both procedures. The difference in the hypotensive effect between the 2 procedures was similar for both procedures (Table 8).

Three patients in group 2 needed two antiglaucoma agents to achieve adequate hypotensive effect and one patient in group 1 needed one drop.

Three eyes in the TM group had hypotension (<6 mmHg) in the early postoperative period but only one of them was taken back to operating theatre and more sutures were placed to the scleral flap. Two patients in the Ex-PRESS group had hypotension but were managed conservatively.

## 4. Discussion

In this study, we investigated the effect of the two antiglaucoma procedures on the corneal biomechanical properties. We found that both indices of the corneal biomechanical properties (CH and CRF) changed significantly in the postoperative period and these changes were similar in both groups. None of the indices returned to the preoperative values 1 year after surgery.

Corneal hysteresis is an indicator of the viscous properties of the cornea, which are due to the presence of glycosaminoglycans, the proteoglycans, and the extracellular matrix [15]. Its value in non-diseased eyes in adult population is around 10.2 mmHg [16]. Regarding its significance in glaucoma, numerous studies agree that CH is lower in eyes with POAG, ocular hypertension, and normal tension glaucoma [17]. Corneal hysteresis was related to glaucomatous field damage progression [18] as well as functional deterioration in the form of reduction of the retinal nerve fiber thickness [19]. In addition to the above, the biomechanical properties of the surface of the eye may reflect similar properties of the lamina cribrosa (LC) of the optic nerve. Lower CH values mean that the cornea (or other tissues like LC) cannot absorb the energy that is exercised on them efficiently and deform at a great extent. On the other hand, tissues with high CH can absorb energy efficiently and do not deform as much. Eyes with lower CH are more prone to glaucomatous damage of the optic nerve as the connective tissue around it deforms significantly (compared to eyes with high CH) as a result of the effect of the IOP and this can lead to damage of optic nerve fibers that run through the LC [20].

Research bears conflicting results regarding the relationship of baseline CH and GAT-IOP. Touboul et al. [21] compared the correlation between GAT-IOP and CH in five groups (normal, glaucoma, keratocus, laser in situ keratomileusis, photorefractive keratectomy) and found a strong correlation only in the glaucoma group. Kaushik et al. [22] analyzed the correlation of CH and GAT-IOP in a cohort comprised of normal subjects, glaucoma suspects, ocular hypertensives, primary angle closure disease, POAG, and normal tension glaucoma patients and found a strong correlation between the two variables. Regarding to the key issue of which device clinicians should use to measure the IOP, Pillunat et al. [23] argue that GAT underestimates the IOP by 3–4 mmHg after trabeculectomy and that IOPcc is a more reliable index of the real IOP. On the other hand, Kaushik et al. [22] believe that the Goldmann tonometer should be used in routine practice. Given the fact the ORA is not widely available in many ophthalmological settings and for an accurate measurement of the IOP and the corneal biomechanical indices to be obtained, a reliable measurement must be taken with the ORA [24], it seems that the Goldmann tonometer is an accurate and at the same time easily accessible instrument. The authors of this study believe that in some cases where the clinical picture of a patient demands an in-depth measurement and evaluation of multiple variables, then in such a case, it would be advisable to use the ORA as well.

The importance of these properties lies in the fact that they affect the measurements of the IOP to a greater effect than the central corneal thickness [25]. As the IOP reduction is the main target of the antiglaucoma operations, it is of paramount significance that the clinician can estimate as accurately as possible the true IOP. The effect of phacoemulsification on the corneal biomechanical properties has been measured in other studies. de Freitas et al. [26] found that the CH temporarily decreases in the immediate postoperative period but returns to the preoperative values later. CRF on the other hand decreased significantly but its values did not return to the preoperative levels at the end of the study period. Zhang et al. [27] also found that CH returned to the preoperative values after a decrease for a short period of time. They did not observe, however, any significant changes of the CRF.

Several other investigators looked at the effect of the antiglaucoma surgeries on the cornea. Sun et al. [28] found that the CH increases in eyes with chronic primary angle closure glaucoma after TM and this tendency remains constant four weeks after surgery. In another study, the influence of TM on the CH and CRF were investigated by Pillunat et al. [23]. Neither of the indices changed significantly postoperatively although there was a trend for reduction of both indices. Interestingly, both IOP measurements given by the ORA (IOPcc and IOPg) were significantly higher than Goldmann tonometry.

However, in TM, surgeons do not insert a drainage device. In this study, we included a group that had an Ex-PRESS implant inserted under a scleral flap in order to achieve IOP reduction. We found one study in which the effects on the cornea of a drainage device were measured. Pakravan et al. [29] compared the effects of TM, combined phacoemulsification-TM (PT), Ahmed drainage device, and phacoemulsification on the cornea. They reported that CH increased in all groups three months postoperatively, while CRF decreased. They speculated that the reasons for the increase of the CH are the reduction of the IOP and the discontinuation of the topical antiglaucoma medication. The latter have been found to cause an increase of the CH [30,31], which does not explain the higher CH values after a successful antiglaucoma procedure.

In order to investigate whether the increase of the biomechanical markers was due to the decrease of the IOP, we performed a regression analysis taking into account the IOP change. According to our data, the change of both indices (before and after surgery) was not correlated to the IOP change (before and after surgery). Multivariate analysis taking into account the GAT-IOP and age did not show any correlation with the biomechanical properties. According to our results, it seems that the glaucoma procedures affect the integrity of the cornea permanently. This is to be expected as less invasive procedures such as uneventful phacoemulsification does alter the corneal structure even if this is only temporary [25]. We would not expect to witness these structural changes at the level of lacrimal cribrosa as the consequences of the glaucoma surgery have local effects and not global.

Our results agree with the results of previous studies in terms of an increase of the CH after antiglaucoma surgery. However, we found that the CRF decreased postoperatively in both groups and this trend remained the same for the entire study period. A similar effect was noted by Pillunat et al. [31] after selective laser trabeculoplasty. CH and CRF measure different properties of the cornea and they are influenced by different factors. CH represents the viscous properties and the CRF the elastic. The reason for the decrement of the CRF values can be the fact that the cornea becomes less elastic after surgery due to effect of the remodeling of the ocular tissues as a response to ocular trauma.

It has been shown that the CH can increase or decrease as the cornea becomes stiffer (old age, cross linking) [32,33]. The role of the CH as an indicator of the corneal stiffness has been debated. There are other factors that influence corneal stiffness (other than viscosity) such as elasticity, hydration, thickness, and extracellular material. However, the increase of the CH after reduction of the IOP (irrespective of its cause) has been shown in many studies [27,28,29,30,31,32]. The alterations of the CRF in our study seem to represent the influence of the surgery and/or the presence of a foreign body (Ex-PRESS mini-shunt) in the vicinity of the cornea.

Our study has several limitations. The small sample size of the two groups is a limiting factor in making safe deductions about the effect of the TM and the Ex-PRESS device on the cornea. We also have not taken into account the IOP as a covariate in the statistical analysis of the CH and CRF. On the other hand, we followed up our patients for a year, which is longer than the follow-up period in similar studies.

## 5. Conclusions

In summary, both surgical techniques have shown to cause an increase of the CH to the same extent postoperatively and a decrease of the CRF. Clinicians should bear in mind these biomechanical changes after an antiglaucoma procedure and adapt their postoperative evaluation and plan accordingly.

## Figures and Tables

**Table 1 jcm-10-00802-t001:** Demographic characteristics and ocular parameters of the two groups preoperatively.

Demographics and Ocular Parameters	Ex-PRESS	Trabeculectomy	*p* Value
Sex (M/F)	10/9	6/5	
Age: range (mean)	16–81(62.4)	60–78 (67.2)	0.18
Diagnosis POAG PXG	11 8	7 4	
Pre-op IOP (mean ± SD) (mmHg)	29.4 ± 7.39	33.2 ± 8.61	0.2
Pre-op antiglaucoma agents (mean ± SD)	2.2 ± 0.7	2.3 ± 0.6	
Pre-op CH (mean ± SD)	7.31 ± 1.13	7.82 ± 2.55	0.24
Pre-op CRF (mean ± SD)	10.25 ± 2.76	11.11 ± 1.99	0.43

CH: Corneal hysteresis, CRF: Corneal resistance factor, Μ: Male, F: Female, IOP: Intraocular pressure, POAG: Primary open angle glaucoma, PXG: Pseudoexfoliation glaucoma, SD: Standard deviation.

**Table 2 jcm-10-00802-t002:** Corneal hysteresis in group 1 (Ex-press group) at different time points.

CH (Mean)	CH	Mean Difference	SE	*p* Value ^a^ (Repeated Measures ANOVA)
Preop: 7.31	Postop—1 month	0.606	0.308	0.394
	—6 months	1.011	0.316	**0.0318**
	—12 months	1.056	0.313	**0.0218**

^a^: *p* value Bonferroni corrected, *p* values < 0.05 are bold, CH: corneal hysteresis, SE: standard error.

**Table 3 jcm-10-00802-t003:** Corneal resistance factor in group 1 (Ex-PRESS group) at different time points.

CRF (Mean)	CRF	Mean Difference	SE	*p* Value ^a^ (Repeated Measures ANOVA)
Preop: 10.26	Postop—1 month	−2.178	0.540	**0.0052**
	—6 months	−2.133	0.482	**0.0022**
	—12 months	−1.9	0.479	**0.0060**

^a^: *p* value Bonferroni corrected, *p* values < 0.05 are bold, CRF: corneal resistance factor.

**Table 4 jcm-10-00802-t004:** Corneal hysteresis in group 2 at different time points.

CH	Median	Minimum–Maximum	CH Preop—Time Points	*p* Value ^a^ (Friedman Test)
preop	7.85	3.9–13.9	Preop	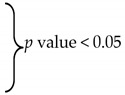
1 month	8.9	6.2–14.7	—1 month	
6 months	8.8	6.4–14	—6 months	
12 months	8.6	6.6–14.8	—12 months	

^a^: Conover post—hoc test, CH: corneal hysteresis.

**Table 5 jcm-10-00802-t005:** Corneal resistance factor in group 2 at different time points.

CRF (Mean)	CRF	Mean Difference	SE	*p* Value ^a^ (Repeated Measures ANOVA)
Preop: 11.11	Preop—1 month	2.9	0.548	0.0015
	—6 months	2.783	0.438	0.0003
	—12 months	2.825	0.422	0.0002

^a^: *p* value Bonferroni corrected, *p* values < 0.05 are highlighted, CRF: corneal resistance factor, SE: standard error.

**Table 6 jcm-10-00802-t006:** Difference of the corneal hysteresis (CH) values (preoperative values to postoperative values at all time points) for the two groups.

CH (Preop-postop Time Points)	1 Month Mean ± SD	*p* Value (Welch Test)	6 Months Mean ± SD	*p* Value (Welch Test)	12 Months Mean ± SD	*p* Value (Welch Test)
Ex-PRESS group	0.75 ± 2.23	0.44	1.59 ± 2.82	0.56	1.64 ± 2.82	0.47
Trab group	1.22 ± 1.08		1.15 ± 1.11		1.1 ± 1.06	

CH: Corneal hysteresis, SD: Standard deviation.

**Table 7 jcm-10-00802-t007:** Difference of the corneal resistance factor (CRF) values (preoperative values to postoperative values at all time points) for the two groups.

CRF (Preop-Postop Time Points)	1 Month Mean ± SD	*p* Value (Unpaired *t*-Test)	6 Months Mean ± SD	*p* Value (Unpaired *t*-Test)	12 Months Mean ± SD	*p* Value (Unpaired *t*-Test)
Ex-PRESS group	–1.67 ± 2.7	0.23	–2.13 ± 2.04	0.62	–1.9 ± 2.03	0.27
Trab group	–2.9 ± 1.89		–1.7 ± 2.7		–2.81 ± 1.47	

CRF: Corneal resistance factor, SD: Standard deviation.

**Table 8 jcm-10-00802-t008:** Mean intraocular pressure (IOP) before and after surgery at the predetermined time points for the two groups.

Mean IOP	Trabeculectomy (Mean ± SD)		ExPRESS (Mean ± SD)		*p* Value (Unpaired *t*-Test)
Preop	33.2 ± 8.61		29.4 ± 7.39		0.2
—1 month	10.4 ± 4.86	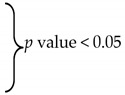	12.9 ± 4.57	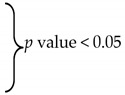	0.16
—6 months	13.6 ± 4.1		15.9 ± 3.23		0.10
—12 months	15.9 ± 3.07		16.1 ± 3.77		0.87

IOP: Intraocular pressure, SD: Standard deviation.

## Data Availability

Authors are willing to share the individual deidentified participant data including written consent forms and study information leaflets for at least one year following the publication of our manuscript, acceptable in print form. Please note that all relevant data is in Greek language.

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
