# Peer review of "The Effect of Antiglaucoma Procedures (Trabeculectomy vs. Ex-PRESS Glaucoma Drainage Implant) on the Corneal Biomechanical Properties"

_jcm, 2021, doi:10.3390/jcm10040802_

Round 1

Reviewer 1 Report

This is an interesting and well written manuscript. 

Minor comments

Line 135: "Medcalc software version 18 was used for the statistical analysis." Please cite the software properly (including full version, country, year) 

Line 138 - 139: ".... when the distribution was not normal." Please rephrase "...when the distribution was skewed".  

Reviewer 2 Report

The authors investigated the changes in corneal hysteresis (CH) and corneal resistance factor (CRF) after glaucoma filtration surgeries. Sample size was too small and multivariate analyses including IOP were not performed. I am afraid that this study showed no novel information.

<Major issues>

  1. The authors described that the participants were included in a consecutive manner (line 87) and the study was conducted over a relatively long period (line 84). However, they included only 30 patients totally. TM group was composed of only 11 eyes. Why was the sample size too small? Furthermore, it is not clear how they decided the procedures, trabeculectomy or ExPRESS. An appropriate explanation is needed.
  2. Several previous reports have investigated the CH and CRF after filtering glaucoma surgery. The results of this study were not so novel. Although it has been well known that CH and CRF are greatly affected by IOP, they did not perform multivariate analyses including IOP measurements.

<Minor issues>

  1. Lines 40-43: “There are several models of the implant but in the current study 40 the P50 model was used. It has an internal diameter of 50μm, an external diameter of 0.4 41 mm and is 2.46 mm long. The use of this mini shunt has shown to be as effective as the 42 standard TM with fewer side effects [7,8].” This should be written in the method section.
  2. Introduction: There are 9 paragraphs in the introduction section, which is too many and too short in each paragraph. It might be better to reorganize them.
  3. Line 152 and Tables 2, 3, and 5: ANOVA should be explained in the methods section. In Tables 2, 3, and 5, the result presentation about ANOVA looks strange for me. Did the authors really perform ANOVA here? What kind of post-hoc analysis was chosen?

Reviewer 3 Report

This is a very interesting study in an area that is relatively new (CH and CRH). I would like more background on the significance of CH and CRH, and also more discussion on why this is clinically relevant. For example, CH is related to the measured IOP by Goldmann applanation. How would that change post-operative management of the patient? Should it? If that is beyond the scope of this paper, then it would still be worth mentioning for future directions. 

I take issue with the presentation of the tables. Tables 1,2, 3 and 4 compare pre-op to month 1, 6 and 12 , and then compares month 1 to pre-op, month 6, and 12, etc. This is highly redundant and makes the paper difficult to read, and I do not see the reason why it is important to show it all. It might be more effective and to the point just to show the comparison between pre-op and all the other time points. Also, I noticed that the authors used a negative sign to indicate that the CH actually increased post-operatively. That might be confusing for the reader (for example, on initial read I interpreted that as a decrease after surgery which contradicted the body of the text). 

Lastly, because CH and CRH are relatively new factors to consider in ophthalmology, the reader might find it helpful to know what the range of normal is, and if the difference between pre-op and post-op periods is significant or not. 

Round 2

Reviewer 2 Report

It has been well known that CH/CRF are greatly affected by IOP. Although I strongly suggested that multivariate analysis including IOP measurements should be performed in my previous review, the authors did not. I understand that the univariate analysis showed no significant correlation between the IOP change and CRF/CH changes in their cases. However, if they want to say that the type of surgical procedure independently affect CRF/CH, multivariate analysis should be needed.

Author Response

We carried out multivariate analysis as correctly noted and the results are discussed in the manuscript (marked in brown)